# Could Lower Testosterone in Older Men Explain Higher COVID-19 Morbidity and Mortalities?

**DOI:** 10.3390/ijms23020935

**Published:** 2022-01-15

**Authors:** Luis M. Montaño, Bettina Sommer, Héctor Solís-Chagoyán, Bianca S. Romero-Martínez, Arnoldo Aquino-Gálvez, Juan C. Gomez-Verjan, Eduardo Calixto, Georgina González-Avila, Edgar Flores-Soto

**Affiliations:** 1Departamento de Farmacología, Facultad de Medicina, Universidad Nacional Autónoma de Mexico, Mexico City 04510, Mexico; lmmr@unam.mx (L.M.M.); biancasromero_@hotmail.com (B.S.R.-M.); 2Laboratorio de Hiperreactividad Bronquial, Instituto Nacional de Enfermedades Respiratorias “Ismael Cosío Villegas”, Mexico City 14080, Mexico; bsommerc@hotmail.com; 3Laboratorio de Neurofarmacología, Instituto Nacional de Psiquiatría “Ramón de la Fuente Muñiz”, Mexico City 14370, Mexico; hecsolch@imp.edu.mx; 4Laboratorio de Biología Molecular, Instituto Nacional de Enfermedades Respiratorias “Ismael Cosío Villegas”, Mexico City 14080, Mexico; aaquino223@gmail.com; 5Dirección de Investigación, Instituto Nacional de Geriatría, Mexico City 10200, Mexico; jverjan@inger.gob.mx; 6Departamento de Neurobiología, Dir. Inv. en Neurociencias, Instituto Nacional de Psiquiatría “Ramón de la Fuente Muñiz”, Mexico City 14370, Mexico; ecalixto@imp.edu.mx; 7Laboratorio de Oncología Biomédica, Instituto Nacional de Enfermedades Respiratorias “Ismael Cosío Villegas”, Mexico City 14080, Mexico; ggonzalezavila22@gmail.com

**Keywords:** testosterone, COVID-19, SARS-CoV-2, viral replication, calcium regulation, aging, inflammaging

## Abstract

The health scourge imposed on humanity by the COVID-19 pandemic seems not to recede. This fact warrants refined and novel ideas analyzing different aspects of the illness. One such aspect is related to the observation that most COVID-19 casualties were older males, a tendency also noticed in the epidemics of SARS-CoV in 2003 and the Middle East respiratory syndrome in 2012. This gender-related difference in the COVID-19 death toll might be directly involved with testosterone (TEST) and its plasmatic concentration in men. TEST has been demonstrated to provide men with anti-inflammatory and immunological advantages. As the plasmatic concentration of this androgen decreases with age, the health benefit it confers also diminishes. Low plasmatic levels of TEST can be determinant in the infection’s outcome and might be related to a dysfunctional cell Ca^2+^ homeostasis. Not only does TEST modulate the activity of diverse proteins that regulate cellular calcium concentrations, but these proteins have also been proven to be necessary for the replication of many viruses. Therefore, we discuss herein how TEST regulates different Ca^2+^-handling proteins in healthy tissues and propose how low TEST concentrations might facilitate the replication of the SARS-CoV-2 virus through the lack of modulation of the mechanisms that regulate intracellular Ca^2+^ concentrations.

## 1. Introduction

Despite the restrictive measures (i.e., isolation, social distancing) and massive vaccination campaigns, the number of people affected by the current COVID-19 pandemic is growing daily. As of 5 January 2022, there have been 295,577,202 confirmed cases of COVID-19, including 5,460,818 deaths, and these numbers are continuously evolving [1]. It is essential to evaluate the current guidelines and strategies in providing safe health services to ensure efficacy in the management of the current pandemic [2]. Global Health 50/50 points out that most data available indicate infection degree is equal for men (49.89%) and women (50.1%) and that no consistent pattern in terms of who is most likely to be diagnosed with COVID-19 exists [3]. This tendency was also reported by the World Health Organization, which shows that there is little difference in the number of confirmed cases in men (49%) and those in women (51%) [4]; not surprisingly, the Mexican population follows the trend: from the total number of confirmed cases (4,008,648), 50.14% corresponds to females and 49.86% to males. The total number of deaths is 299,711 [1,5]. According to the Center for Systems Science and Engineering (CSSE) at Johns Hopkins University, in January 2022, confirmed cases in America were 105,416, 916 and USA had the highest incidence [1]. Global Health 50/50 reports that in this country male patients between 50 and 64 years of age presented a death toll almost two times higher than in women of the same age (293.26 vs. 170.66 per 100,000, respectively) [3]. Indeed, it has been observed that most COVID-19 fatalities were older males, even in those countries with a higher number of confirmed cases in women. Seemingly, once infected, men are at a higher risk of dying from COVID-19 than women, and this risk directly correlates with age (Figure 1) [3,6,7].

The former fact has been noticed frequently since men with coronavirus infections have shown a lower survival rate than women. In the SARS-CoV epidemics of 2003 and the Middle East respiratory syndrome epidemics of 2012, men had substantially higher fatality rates than women, as in the current COVID-19 pandemic [8]. This gender-related difference in COVID-19 infection susceptibility, severity, and mortality has not been thoroughly explained, although it has been proposed, it might be attributed to genetic, immunological, and hormonal differences. Among these possibilities, the latter seems adequate to explain at least partially the gender-related observations. Furthermore, since the hormone steroid, testosterone (TEST) plasmatic concentrations decrease with aging and the presence of comorbidities (obesity, diabetes mellitus, and cardiovascular diseases) increases during the same period, both circumstances might worsen SARS-CoV-2 patients’ prognosis (Figure 1) [9,10,11,12]. Understandably, multiple studies have been carried out trying to predict the outcome of the disease in patients. In SARS-CoV-2 infected men, TEST has been reported to exert immunosuppressive effects [13] and modulate inflammation [14], which may contribute to attenuated antibody response and worsen the prognosis in comparison to women [14]. SARS-CoV-2 viral entry to host cells has been reported to be through the interaction of the viral spike protein (S) and the Angiotensin-Converting Enzyme 2 (ACE2) receptor, facilitated by the Type II Transmembrane Serine Protease (TMPRSS2) priming the S protein [15,16]. Male sex hormones are also believed to increase the expression of the ACE2 receptor, favoring the SARS-CoV-2 viral infectivity [15]. Furthermore, androgens, including TEST, are the only known promoters of the expression of TPMRSS2 through the activation of the androgen receptor (AR) [17,18]. Not only might TEST participate in the physiopathology of SARS-CoV-2, but the virus can interfere in the hormone’s production [19,20,21,22]. However, when comparing young adult men with elderly patients, when TEST concentrations are progressively decreasing, we observe a greater severity and mortality; therefore, the above mentioned TEST immunosuppressive effects in COVID-19 patients might not be justified.

In this context, we propose that low plasmatic levels of TEST can be determinant in the infection’s outcome and the replication of the SARS-CoV-2 virus through the modulation of the mechanisms that regulate intracellular Ca^2+^ concentrations ([Ca^2+^]i) in host cells. In this regard, it has been reported that dysfunctional [Ca^2+^]i homeostasis mechanisms are necessary for the replication of certain viruses, such as influenza A virus (LVA) [23], Japanese encephalitis virus (JEV), Zika virus (ZIKV), dengue virus (DENV), and West Nile virus (WNV) [23,24,25]. Meanwhile, it was recently published that the over-activation of the ryanodine receptor (RyR) channel [26] and the voltage-dependent Ca^2+^ channel (VDCC) deregulate [Ca^2+^]i homeostasis playing an essential role in SARS-CoV-2 infection and cell replication [27]. It is not yet determined whether other cellular mechanisms are affected by SARS-CoV-2 infection and viral spread (Figure 2).

On the other hand, TEST modulates the activity of diverse proteins that regulate calcium homeostasis and its signaling. For instance, it has been reported that, in different systems, it blocks L-type voltage-dependent Ca^2+^ channels (L-VDCC), store-operated Ca^2+^ channels (SOCCs), transient receptor potential (TRP) channels, inositol 1,4,5-triphosphate receptors (IP_3_R) and promotes prostaglandin E2 (PGE2) [28,29,30,31], contributing to maintaining the basal intracellular Ca^2+^ concentration (b[Ca^2+^]i) and favoring the tissues basal functions (Figure 2) [28,29,30,31,32]. It is important to emphasize that these mechanisms are found in almost all tissues and cells of the body.

In summary, a wide range of evidence from different cell types points out that TEST interacts with various essential regulatory proteins that maintain [Ca^2+^]i homeostasis. Even though this androgen’s physiological role has not been fully elucidated, some evidence hints at its detrimental role in COVID-19 patients warranting further research to understand better the possible effects that TEST could have on the infection and replication of the SARS-CoV-2 virus.

Nevertheless, age is the principal risk factor associated with an increase in severity and mortality in COVID-19 patients [9]. One contributing factor that could explain this matter is “inflammaging”, a chronic inflammatory state observed in the elderly [9]. The decline in TEST levels is associated with age and can participate in the regulation of inflammaging in men [9]. Seemingly, TEST declining plasmatic concentrations in the older men could provide essential hints on the role of this androgen in the pathophysiology of COVID-19 patients.

Because of the above-described issues, we propose herein the following points: (1) SARS-CoV-2 replication depends on [Ca^2+^]i handling proteins; (2) TEST promotes calcium homeostasis at normal plasmatic concentrations; (3) Diminished plasmatic TEST concentrations dysregulate calcium homeostasis; finally, (4) TEST deficiency enhances inflammaging that exacerbates SARS-CoV-2 pathophysiology.

## 2. Calcium Signaling

The calcium ion (Ca^2+^) is a versatile second messenger in all cell types and regulates multiple signaling processes responsible for essential cell functions. The processes it can regulate are time-dependent: in microseconds, exocytosis is generated, in milliseconds, it initiates contraction, and in minutes or hours, it originates events, such as fertilization, proliferation, transcription, gene regulation, and apoptosis [33,34]. Under resting conditions, cells maintain cytosolic Ca^2+^ concentrations ranging from 100 nM to 150 nM [35]; exquisitely regulated mechanisms maintain the equilibrium between the extracellular milieu (Ca^2+^ concentrations ~2 mM) and intracellular Ca^2+^ stores (Ca^2+^ concentrations ~5–10 mΜ) [36,37]. The homeostasis in Ca^2+^ signaling is determined by a balance between the proteins that increase Ca^2+^ within the cytoplasm: L-VDCC, SOCCs, receptor-operated Ca^2+^ channels (ROCCs), Na^+^/Ca^2+^ exchanger in its reverse form (NCX_REV_), IP_3_ receptor (IP_3_R), and ryanodine receptor (RyR) and proteins that decrease concentrations to basal levels: plasma membrane Ca^2+^ ATPase (PMCA), sarcoplasmic reticulum Ca^2+^ ATPase (SERCA), Na^+^/Ca^2+^ exchanger (NCX) and the mitochondrial uniporter [36,38,39,40] Importantly, alterations in this Ca^2+^-dependent homeostatic mechanism might participate in various pathophysiological conditions, including viral infections [33].

## 3. Viral Modifications of Host Cell Calcium Homeostasis

Viruses are intracellular invasive particles that exploit the host cell’s machinery to propagate the viral lifecycle; particularly the intracellular Ca^2+^ signaling system is hijacked during viral entry, viral gene replication, virion maturation, and release of various viral species [23,41]. Dysfunctions in the host cell’s Ca^2+^ apparatus have been reported during a viral infection, leading to abnormal [Ca^2+^]i [23].

One of the primary viral targets of the Ca^2+^ apparatus are the VDCCs. Some studies have shown that the Ca_v_1.2 channel serves as a receptor of the influenza A virus (IAV) and is necessary for its entry into the host cell [23]. This is further supported by the inhibition of IAV infection when VDCC blockers are used, such as verapamil [23,42]. VDCC blockers have also been shown to effectively inhibit the infection of the West Nile virus (NWV) and severe fever with thrombocytopenia syndrome virus (SFTSV) by inhibiting the virus-cell fusion step [23,43,44]. The increase in [Ca^2+^]i produced by certain viruses through VDCCs has also been demonstrated to be necessary for viral replication, and VDCC blockers have proven to be effective antiviral agents against Japanese encephalitis virus (JEV), ZIKV, DENV, and WNV [23,24,45]. It has been shown that Ca^2+^ binds to the fusion protein (FP) of MERS-CoV and the 2 FP domains on the S protein of SARS-CoV during the entry stage of both virus types [27,46,47], nevertheless, this phenomenon still requires further investigation. The use of VDCC blockers has also been associated with lower mortality and decreased risk for intubation in COVID-19 patients; therefore, Ca^2+^ could also potentially be involved in the viral entry stage (Figure 2) [48].

It was recently published that over-activation of the RyR channel and the associated alteration of [Ca^2+^]i homeostasis play an essential role in SARS-CoV-2 infection and intracellular replication. The TGF-β signaling pathway over-activation by this virus and reactive oxygen species (ROS) production leads to Ca^2+^ leak from RyR channels in the sarcoplasmic reticulum (SR). This effect is produced through oxidation and protein kinase A (PKA) phosphorylation, uncoupling the regulating protein calstabin (FKBP12.6) from the RyR channel, destabilizing the closed state and favoring an open state [49]. Another mechanism proposed for RyR channel dysfunction during SARS-CoV-2 infection is through cathepsin L13 (a protease expressed in the host cell’s plasma membrane); this enzyme has also been shown to participate in the over-activation of RyR channels, promoting a leaky state. The increase in [Ca^2+^]i also favors cathepsin L13 activity that allows the viral entry through the cleavage and activation of the S protein. The increase in intracellular Ca^2+^ also promotes the release of the virus from the endosome into the host cell [26,50].

Ca^2+^ release from the SR can also be triggered through activation of the IP_3_R, and it is a known target for some viruses during the early stages of viral infection to promote replication. The human cytomegalovirus (HCMV) interacts with P2Y_2_ purinergic receptors to increase the production of IP_3_ [51,52], while the human immunodeficiency virus (HIV) upregulates intracellular IP_3_ [53] and the human T-cell lymphotropic virus type 1 (HTLV-1) directly activates the IP_3_R (Figure 2) [54].

The final stage of the viral lifecycle consists of the extracellular release via exocytosis from the host cell, also called budding; in four hemorrhagic fever viruses, the STIM1/Orai1- mediated Ca^2+^ release is essential for this step [23,47,55]. This was also demonstrated when DENV yield was significantly reduced by SOCCs antagonists [56]. The influx of Ca^2+^ through SOCCs is a particular hallmark of rotavirus infection, and the mechanism for this action has been established to be through the activity of a nonstructural protein 4 (NSP4), a viroporin acting as an ion channel in the SR (Figure 2) [57,58].

In order to maintain proper intracellular Ca^2+^ homeostasis, calcium pumps and exchangers are required to decrease [Ca^2+^]i, namely PMCA, SERCA, and NCX [33]. The disruption of any of these proteins would increase [Ca^2+^]i, a phenomenon that has been implicated in different stages of the viral cycle [25,59]. The participation of SERCA in the viral genome replication stage was demonstrated through a SERCA inhibitor that showed antiviral activity against respiratory syncytial virus (RSV) strains [60]. In AIDS transgenic mice that express replication-incompetent HIV-1, cardiac dysfunction has been linked to increased SERCA2 expression [61]. On the other hand, rotavirus infection activates NCX in its reverse mode (where one Ca^2+^ enters the cytosol and three Na^+^ ions are expelled) mediated by NSP4 (Figure 2) [62].

Interestingly, studies in structural homology, bioinformatics and metanalyses suggest that Ca^2+^ might participate in SARS-CoV-2 entry into host cells [25]. Furthermore, this is supported by studies showing that VDCC blockers inhibit viral cell entry [26,27]. Although further research is required to understand the extend of Ca^2+^ participation in SARS-CoV-2 pathogenesis, Ca^2+^ handling proteins could be a potential target in treating COVID-19 patients.

As the COVID-19 infection progresses, why men present disproportionately higher infection and mortality rates remain unclear. As of yet, no evidence links directly TEST and this higher susceptibility in men. Because it is well known that TEST physiologically participates in regulating Ca^2+^ handling proteins activity, these effects might help elucidate the paradigm concerning the relationship between TEST and COVID-19 severity and mortality in males. Conceivably, this androgen’s plasmatic concentrations might correlate with COVID-19 severity, i.e., lower concentrations worsen the prognosis, particularly in older men.

## 4. Role of COVID-19 in Testosterone Production

As stated before, SARS-CoV-2 enters the cell through the ACE2-S protein complex; therefore, the targeted cells are those that express ACE2. This protein has been implicated in regulating two testicular functions: steroidogenesis and spermatogenesis and is expressed in four testicular cells: seminiferous duct cells, spermatogonia, Leydig cells, and Sertoli cells [63,64,65,66]. The expression of ACE2 seems to be linked with age, having a higher expression in younger men and indicating a high risk for potential infections of the testis in this population [63,65,67]. SARS-CoV-2 in semen samples and testicular biopsies of patients with COVID-19 were investigated. Interestingly, only two studies found the virus in semen [19,68,69], contradicting other reports that did not [20,21,22,70,71,72,73,74]. Indeed, more studies are needed to further clarify this issue.

Urogenital infections are known risk factors for male infertility, mainly due to the impact of inflammation on reproductive function [63,75,76]. Cytokines are known regulators of male reproduction system health [63,75], and local production in testis has been described [75,77]. Then again, the cytokine storm is a characteristic trait of COVID-19 infection, and, as one of its consequences, increased levels of seminal IL-6, TNF-α, and MCP-1 have been described [78]. COVID-19 can affect the proper testicular function and alter TEST production and male fertility, whether transiently or with more permanent implications. Additionally, fever, a prominent symptom in various infectious diseases including COVID-19, is linked with variations in semen and transient decline in male fertility [79,80]. Either by direct harm to the testicular cells, the SARS-CoV-2 virus entering the testicular cells, or the indirect consequence of the inflammatory response, evidence exists that COVID-19 might compromise male fertility. This illness´ effects on testicular function are observed in spermatogenesis alteration and testosterone production. Sperm quality in COVID-19 patients is altered, with a lower percentage of normal sperm morphology and count, and orchitis has been observed in some COVID-19 patients [20,21,22,78].

Various studies in COVID-19 male patients report low levels of circulating TEST, with most cases showing a normalization of TEST levels post-infection, although in approximately 50% of them not reaching standard levels during a 7-month follow-up, and up to 10% decreasing even further [20,81,82]. Long-term health implications of COVID-19 infection are still unknown, and male infertility as a possible sequel is investigated, especially with the rising interest in the chronic consequences that COVID-19 might pose. Such is the case of the emerging “long-haul”, a term used to describe the clinical duration of symptoms extending past the acute and post-acute infection, generally lasting around 28 days [83,84,85].

Severe COVID-19 cases are associated with impaired viral control and higher viral RNA load [86,87]. Interestingly, lower TEST levels have been found to correlate with COVID-19 severity [88], probably related to the fact that TEST could be implicated in viral replication regulation. COVID-19 adverse effects on TEST production will induce a worse infection.

## 5. Testosterone’s Modes of Action at the Cellular Level

TEST binds to membrane-bound or nuclear receptors and triggers genomic (classical) effects that occur after a long period (hours to days). Meanwhile, nongenomic (non-classical) effects occur in a short period (seconds to minutes) and are independent of the androgen receptor (AR) occupancy by the male sex steroids [28,89]. The AR is present in almost all tissues and cell types, including the brain, heart, lung, and immune system cells [28,89,90].

TEST plays essential roles in Ca^2+^ homeostasis in several muscles, i.e., airway, cardiac and vascular. For instance, in airway smooth muscle (ASM), this androgen has benefic effects and seems to participate in the sexual dimorphism observed in many respiratory diseases, such as asthma that shows lower incidence in adult males than in females, as does symptom severity [91,92]. Some studies also point out that a single high dose of exogenous TEST induces significant bronchodilation [93], the therapeutic potential of this androgen that deserves further investigation.

In ASM, TEST tissular effects are related to the regulation of intracellular Ca^2+^ levels. We recently found that in this tissue, TEST inhibits L-VDCC and SOCCs [28,29,30]. Additionally, at physiological concentrations (nM, nmol/L), TEST induced a decrease in [Ca^2+^]i through the phospholipase C-β/inositol 1,4,5-trisphosphate (PLC_β_/IP_3_) signaling pathway, by blocking the IP_3_R [31]. Also, in guinea pig ASM, TEST diminishes tone and [Ca^2+^]i. These effects seem to occur by blocking L-VDCC and a constitutively active TRPC3 channel, and probably by PGE2 biosynthesis [28,29,30,31]. These mechanisms also favor ASM basal tone by keeping basal intracellular Ca^2+^ concentration (b[Ca^2+^]i) in unstimulated tissues and by inducing relaxation in tissues pre-contracted with carbachol (CCh) or antigenic challenge (Figure 2) [93].

Additionally, we found that ASM chronic exposure to nanomolar concentrations of TEST induces β_2_ adrenergic receptor expression, hence improving the salbutamol-induced relaxation [94]. This finding was further characterized by patch clamp studies that showed increases in the salbutamol-induced K^+^ currents (IK^+^); this rise was abolished when protein synthesis or transcription inhibitors were used during the TEST chronic exposure [94]. The increase in IK^+^ induces ASM hyperpolarization diminishing the Ca^2+^ entry through voltage dependent channels, and therefore, contributing to keeping lower [Ca^2+^]i.

Many studies have established TEST´s paramount role in immunity and inflammation. Hence, it has been demonstrated that TEST negatively regulates type 2 inflammation and the expression of IL-17A [95,96]. Furthermore, in human ASM, androgens diminish the intracellular Ca^2+^ increment induced by pro-inflammatory cytokines, such as tumor necrosis factor alpha (TNF-α) or interleukin-13 (IL-13) by a genomic effect [97]. All these effects diminish airway hyperresponsiveness and favor a milder asthmatic phenotype. Even though ACE2 expression in human ASM was just recently defined [98] and the entry of SARS-CoV-2 through its association has not been demonstrated yet, it is reasonable to propose that the above-described mechanisms could also be relevant in the SARS-CoV-2 infected males (Figure 3).

The role of sex hormones has been extensively studied in physiological and pathological settings. Low levels of circulating TEST are associated with an increased cardiovascular risk by leading to an increase in inflammation, impaired metabolism, and mitochondrial dysfunction [99,100].

In male rodents, gonadectomy (GDX) reduced the expression of L-VDCC in the heart [101,102,103], and chronic exposure to dihydrotestosterone (DHT) increased the expression of Ca_v_1.2 and peak I_Ca-L_ (L-type Ca^2+^ current) in human ventricular myocytes [103,104]. The NCX protein has also been explored, though the evidence is contradictory. Some studies report that after 2–10 weeks of GDX the expression and activity of NCX were unchanged [103,105,106,107]. In other works, there is evidence that after 2 or 16 weeks of GDX, a decrease in levels of mRNA of NCX occurs and that it could be reversed with supplementation of TEST [101,102,103]. At the moment, the effects of TEST in the regulation of NCX in cardiomyocytes are still unclear and require further investigation (Figure 2).

The administration of TEST at supraphysiological levels for two weeks appears to have a protective effect against myocardial ischemia-reperfusion injury, demonstrating an improvement in functional recovery compared to GDX and placebo groups [105]. The effect was partly attributed to the impact of TEST on [Ca^2+^]i, reducing the end-ischemic [Ca^2+^]i and having a decreased [Ca^2+^]i overload in the postischemic period [105]. Worth mentioning is the [Ca^2+^]i homeostasis in contractile failure, the possibility of developing arrhythmias, and myocyte injury [105,108]. Although the protective effect of TEST in reperfusion injury is evident and is associated with [Ca^2+^]i handling, the effect cannot be attributed to a difference in protein expression of phospholamban (PLB), the NCX, RyR2, or SERCA2a. Yet, the possibility of changes in phosphorylation in any of these proteins remains [105,106,107,108,109]. Similarly, in another study, TEST did not alter protein expression of SERCA, its modulating components sarcolipin and heat shock protein 20 or NCX. However, in GDX rats, the phosphorylated Thr17 and Ser16 forms of PLB were significantly decreased, modulating SERCA activity [103,110,111]. Even though GDX does not modify the levels of expression of RyR, RyR-mediated Ca^2+^ release is decreased after GDX [103,104,105,106], with chronic testosterone exposure (24–30 h) increasing the amplitude of Ca^2+^ sparks [103,104]. The increase in SR Ca^2+^ release from individual Ca^2+^ sparks could be caused by an increase in SR Ca^2+^ content with exposure to TEST [103,104,105,106]; this increase in Ca^2+^ content is attributed to phosphorylation of PLB (Figure 2) [103,110,111].

COVID-19 patients appear to have cardiac dysfunction, leading to cardiac injury, with several studies demonstrating it through cardiac marker elevation and electrocardiogram (ECG) changes [112,113,114,115]. The incidence of cardiac injury is reported to be between 7.2% and 28%, but in severe and critical care patients, the incidence can be between 22% and 44% [112,113,114,115,116,117,118]. Arrhythmias can be a common symptom in COVID-19 patients, requiring close monitoring since they indicate myocardial injury associated with an unfavorable outcome. The incidence of arrhythmias has been reported to be between 17% and 24%, linked with intensive care unit (ICU) admission and death, exacerbating previously known cardiac comorbidities and unfortunately developing in patients without prior history of heart disease [112,115,119,120].

Cardiac arrhythmias could be caused by various factors present in COVID-19 patients, such as hypoxia, pro-inflammatory cytokines, direct myocardial injury, fever, electrolyte imbalances, plaque rupture, hypercoagulability, or many of the medications used to treat COVID-19 patients [115,121]. Concerning the induction of cardiac injury, SARS-CoV-2 has been shown to directly infect cardiomyocytes through internalization of the virus when the viral S protein binds to ACE2, aided by the TPMRSS2 [115,122,123]. Fever, a symptom often present in COVID-19 patients, has also been shown to trigger ventricular arrhythmias, especially in patients with underlying cardiomyopathies [115,124,125,126]. Specifically, pro-inflammatory cytokines can promote an arrhythmogenic state. In COVID-19 patients, some cytokine concentrations are elevated, such as IL-6, IL-1β, IL-2, IL-8, IL-17, G-CSF, GM-CSF, IP10, MCP1, CCL3 and TNF-α; all could lead to the generation of arrhythmias [113,115,127,128,129]. The acute administration of IL-6 increases L-type Ca^2+^ currents (I_CaL_) in ventricular cardiomyocytes [115,130], and in chronic exposure, IL-6 has significantly down-regulated the expression of SERCA2 in ventricular myocytes [115,131]. Additionally, TNF-α reduces I_CaL_ and the expression of SERCA2a by increasing DNA methyltransferase levels, thus enhancing the methylation of its promoter region [115,126,132]. Furthermore, IL-1β has been shown to promote Ca^2+^ spark frequency [115,133]. Although the extent to which TEST plays a role in cardiomyocyte injury during SARS-CoV-2 infections remains uncertain, there exists evidence suggesting it could have a protective role and warrants further investigation.

Studies have demonstrated that TEST affects the cardiovascular system in health and disease. TEST may serve different functions in normal physiological conditions compared with pathophysiological states [134]. TEST concentrations in men remain relatively constant through the reproductive lifetime, in the range of 6–50 nM, and can influence the cardiovascular system functions, regulating vascular resistance, cardiac electrophysiology, and cardiac output, and TEST deficiency may contribute to developing hypertension [135,136,137,138]. Several epidemiological studies have shown an association of low testosterone with cardiovascular disease and conditions, such as metabolic syndrome and type 2 diabetes, which have increased cardiovascular risk [139,140].

It has been reported that TEST exhibits vasodilatory actions, both through acute and chronic mechanisms, and this effect can be observed in different species, including humans, and be reproduced in many vasculature types, i.e., thoracic, coronary, mesenteric, pulmonary, mammary, radial, and umbilical arteries [141,142,143,144,145,146,147,148,149,150,151,152,153,154,155,156].

Many of the mechanisms responsible for producing vasodilation have been deciphered and will be addressed below. One of the best-described mechanisms is acute TEST inhibition of the VDCCs. This effect can be obtained in various models, including rat aorta [157,158,159,160], porcine and rat coronary arteries [147,161,162], rat pulmonary artery [148,151], canine basilary artery [163], human umbilical artery (HUA) [164], and small porcine arteries [165], and can even potentiate the effect of nifedipine [166]. Moreover, TEST can regulate other Ca^2+^ handling proteins that participate in the vasodilatory effect, including the inhibition of ROCCs, which can be observed in rat aorta [157], porcine arteries [164,165], and HUA [164]. Similarly, SOCCs inhibition is observed in rat coronary, pulmonary and aorta arteries (Figure 2) [147,162].

In COVID-19, hypertension has been described as a morbidity risk factor and poor outcome [167]. As an essential vasodilator, TEST can mitigate the risk of hypertension, and its deficiency is linked to increased cardiovascular risk [140]. Moreover, ACE2 inhibitors and ARB (Angiotensin II receptor blocker) administration, two of the primary drug groups used in the treatment of hypertension, have shown to increase the expression of ACE2 [168,169,170]. This overexpression of ACE2 can increase the risk for potential infection by SARS-CoV-2. Therefore, TEST can indirectly mitigate the impact of COVID-19 by decreasing the cardiovascular risk or by lessening the necessity for ACE2 inhibitors or ARBs. The extent of hypertension’s impact on the pathophysiology of COVID-19 is undoubtedly complex and possibly related to underlying comorbidities; this interesting fact remains a guideline for future studies.

In summary, the modulatory effects that physiological concentrations of TEST excerpt on the Ca^2+^ handling mechanisms that participate in the viral lifecycle could lessen the potential infection of SARS-CoV-2. Contrastingly, TEST deficiency has been shown to worsen comorbidities that pose a risk for COVID-19 severity and outcome, including those in the respiratory and cardiovascular systems. TEST plasmatic concentrations decrease with age, and therefore, might constitute a dominant risk factor observed to impact COVID-19 severity and mortality [168,169]. Besides, one of the primary hallmarks of aging is the so-called inflammaging, which also augments the risk of acquiring COVID-19.

## 6. Role of Inflammaging in the Pathogenesis of COVID

Young adults with COVID-19 and a favorable natural course of the disease, present a balance between the ratio of pro-inflammatory and anti-inflammatory cytokines, capable of modulating immune activity and reducing the response at the indicated time. A dysregulation of the immune response, as the chronic state of inflammation known as “inflammaging” in elderly patients, may contribute to the pathophysiology of SARS-CoV-2 [171,172]. Inflammaging has been associated with various pathologies, such as insulin resistance, type 2 diabetes mellitus, cardiovascular disease, Alzheimer’s disease, and cancer [171,172]. Old age is characterized by this chronic state of inflammaging, in which a systemic increase in IL-6, IL-8, TNF-α, IL-13, IFN-γ, and acute phase proteins has been detected, and includes a series of systemic alterations, especially of the immune system. The sum of these factors could favor viral infections as a result of alterations in autophagy and mitophagy activity, increased ROS production, cellular senescence that contributes to the pro-inflammatory profile related to aging, senescence of immune system cells, alteration of the expression of TLRs (toll-like receptors) and decrease in the concentration of vitamin D (Figure 3) [171,172].

Aging is associated with an increase in ROS production, which promotes the pro-inflammatory state through the synthesis of cytokines and the activation of transcription factors including human polynucleotide phosphorylase (hPNPaseold-35), NF-κB, activator protein 1 (AP-1), specificity protein 1 (Sp1), and peroxisomal proliferator-activated receptors (PPARs) [171]. One of the processes responsible for mitigating ROS production is mitochondrial autophagy, known as mitophagy. Autophagy is a catabolic exchange pathway in which dysfunctional or damaged cellular material is degraded; an alteration or decrease in this pathway has been associated with various pathologies characteristic of aging. When autophagic activity declines, it leads to an increase in ROS production. The lower autophagic activity and the enhanced ROS production lead to the activation of NOD-like receptors (NLR), especially NLRP3. As products of the activation of the NLRP3 receptor, the cytokines IL-1β and IL-18 also activate pyroptosis, a form of programmed cell death in which they release their pro-inflammatory cytosolic content to the extracellular space. There is an increase in the proportion of senescent cells in old age, these are characterized by having decreased cell viability and being more susceptible to cellular damage by ROS, and they can also produce cytokines, such as IL-1α, IL-1β, IL-6, IL-8, IL-18, CCL-2, TNF-α, GM-CSF, growth-regulated oncogene (GRO), MCP-2, MCP-3, MMP-1 and MMP-3 [171]. Above all, senescent adipocytes play an essential role in inflammaging. In old age, a redistribution of adipose tissue can be observed, with a decrement in subcutaneous regions and increases in the visceral areas; this could also be altered in age-related diseases, such as sarcopenia. This redistribution is associated with a dysfunction of adipose tissue, an increase in the production of adipokines and cytokines (especially IL-6 and TNF-α), metabolic dysfunction, and predisposes subjects to increased morbidity and mortality from several causes (Figure 3) [173].

Furthermore, immunosenescence also contributes to increasing the progressive loss of all immune effectors in both the innate and cellular immune systems [171,172]. An augmented activation and maturation of dendritic cells (DCs) by cytokines has been reported in this context. It has been described that the T cell population also undergoes essential changes that do not include decreases in cellular counts. There is a poor T cell mitogenic response, an alteration in the CD4+/CD8+ T cells ratio, a reduction of immature T cells, an increase in memory T cells, and the Th17/Treg cells ratio [171]. Macrophages show lower production of specific factors, for instance, fibroblast growth factor, vascular endothelial growth factor, epithelial growth factor, TGFß, toxic free radicals, and nitric oxide synthase expression, and a decrease in phagocytic and chemotactic activity. Lower production of antibodies and their protective effectiveness have been observed within the alterations in the B cell population, corresponding with the mitigated response of specific antigen antibodies, observed in old mice [171]. The changes observed by immunosenescence in older adults produce a chronic inflammatory profile, causing higher age-related morbidity and mortality in COVID -19 (Figure 3) [171,172].

Old age is also accompanied by vitamin D deficiency associated with several chronic degenerative diseases. The non-classical activities of this vitamin are related to immunoregulatory effects. In conjunction with its vitamin D receptor (VDR), it increases macrophages´ autophagic activity and the generation of antimicrobial products and favors a decrease in the expression of pro-inflammatory cytokine genes. These genes are silenced by higher glutathione levels, lowering ROS and suppressing the expression of NF-κB and p38 MAP kinase. Conceivably, elderly patients faced with SARS-CoV-2 infection would be unable to efficiently modulate the inflammatory response, most probably presenting an exacerbated response and severe tissue damage (Figure 3) [171].

A steady decline in TEST plasmatic concentration is associated with age, typically referred to as andropause, and is currently considered late-onset hypogonadism (LOH) [174]. This lowering of TEST plasmatic levels can have clinical repercussions and has been observed to decrease bone mineral density and lean body mass and increase the risk of metabolic syndrome and cardiovascular diseases [11,174,175]. The higher ROS production observed in inflammaging could also contribute to TEST deficiency, since high levels of ROS have been shown to disrupt the male reproductive hormonal profile: directly through oxidative stress and indirectly by acting on the hypothalamic axes of hormone release, decreasing luteinizing hormone (LH) secretion [176,177,178,179]. The treatment with TEST could be beneficial in hypogonadism, particularly in LOH, protecting against the effects of ROS on TEST production. The treatment with low doses of TEST has demonstrated a diminished ROS production in Leydig cells, preventing oxidative damage and upregulating the expression of the steroidogenic acute regulatory protein (StAR), which acts as the limiting-step enzyme in steroidogenesis, resulting in higher TEST synthesis and secretion (Figure 3) [180].

Similarly, TEST replacement therapy attenuated cognitive decline in rats by decreasing oxidative stress damage [181]. TEST production in Leydig cells depends on autophagy; another characteristic of inflammaging is altered autophagy, and disruption, especially in this site with high activity, and could lead to LOH [182]. Additionally, the senescent adipocytes observed in inflammaging can contribute to male infertility. Adipose tissue-mediated inflammation and oxidative stress in obese men can negatively impact TEST production and sperm quality, promoting LOH (Figure 3) [183,184].

Another aspect that has gained interest is the synchronicity between the decline of TEST plasmatic concentration and the development of a pro-inflammatory state [100]. In male diabetic patients, low levels of TEST are associated with a pro-inflammatory condition characterized by high TNF-α concentrations, an impaired metabolic profile, and mitochondrial dysfunction, leading to an increase in cardiovascular risk [99]. TEST deficiency has also been shown to increase IL-6 production in the bone marrow of young mice [9,185]. Inversely, TEST supplementation treatment could prove beneficial in reversing some of the detrimental immunological effects related to age, such as immunosenescence. TEST treatment can decrease the production of IL-6 and other pro-inflammatory cytokines in vitro and in vivo [9,186]. TEST treatment in men with hypogonadism significantly reduced the production of TNF-α and IL-1β and incremented the production of IL-10 (Figure 3) [9,187].

Similarly, in rat autoimmune orchitis, TEST treatment decreased CD4+ T cells, increased Treg cells, and decreased Th1 cytokine production (IFN-γ and IL-2) and other pro-inflammatory cytokines (MCP-1, TNF-α, IL-6) [9,188]. Moreover, the alterations in Th2 response related to aging could benefit from TEST modulation. TEST, through the AR activation, has been shown to suppress Th2-mediated inflammation indirectly by suppressing IL-4 production induced by allergen exposure in mice models [96]. Therefore, LOH could reasonably exacerbate the repercussions that inflammaging could have in the pathogenesis of COVID-19, and it would be interesting to investigate if TEST administration could be beneficial in older men suffering this illness (Figure 3).

At this point, it is important to distinguish between the chronological age (age measured in years from the date you are born to the present) and the biological age (age referred to different physiological and molecular processes, usually measured with distinct biological biomarkers, such as DNA methylation). This distinction may help us to better understand why the COVID-19 pandemic showed to be more lethal on subjects with several comorbidities, such as obesity, diabetes, or hypertension, most of which have shown an acceleration of age (residuals between chronological age estimation and biological age) [189,190,191].

In this sense, several studies have pointed out that biological age is strongly associated with the severity of the disease rather than with the calendar age. Moreover, in a recent article by Chiang-Ling et al., phenotypic age (PhenoAge) measured with several biomarkers and a machine-learning model [192], showed to be associated with severity of COVID-19 when data from the UK Biobank were combined with COVID-19 diagnoses of the UK National Health Service [193]. In this sense, a recent article reported that men have accelerated biological aging during quarantine. Interestingly, this study found that biopsychological age might determine the risk to develop severe COVID-19 [194].

## 7. Conclusions

The higher severity and mortality observed in male COVID-19 patients could be linked to lower TEST protective effects. Illness severity has been associated with TEST deficiency, especially in elder patients. TEST might be modulating SARS-CoV-2 pathophysiology directly (regulating the viral life cycle) and indirectly (mitigating the exaggerated immunological response). The viral hijacking of the Ca^2+^ handling proteins might be a potential target for pharmacological treatment, and the modulatory actions of TEST over these mechanisms could prevent their viral-induced dysfunction. Further research on how low TEST plasmatic concentrations in elderly patients worsen SARS-CoV-2 symptoms is clearly needed.

## Figures and Tables

**Figure 1 ijms-23-00935-f001:**
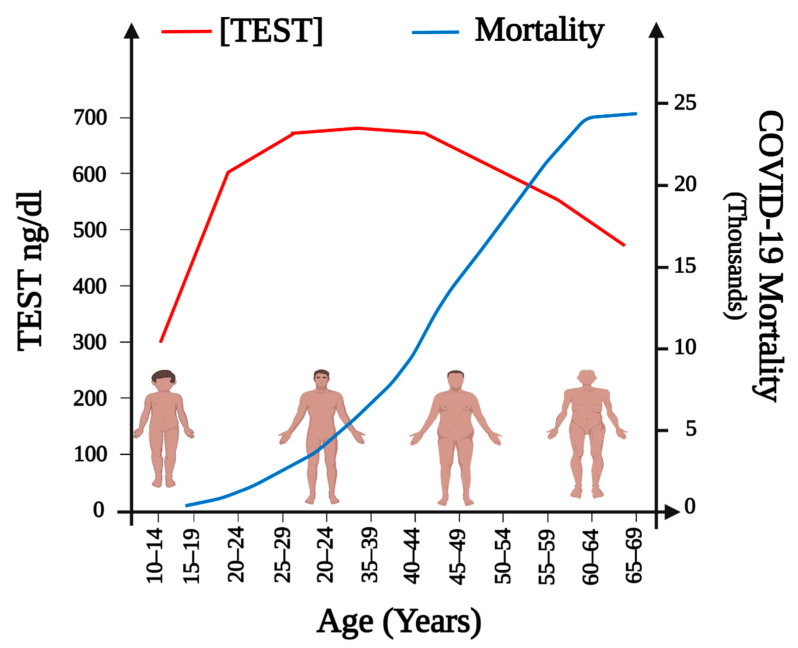
Association of TEST plasmatic concentrations and COVID-19 mortality in Mexican men by age group. Diminished TEST plasmatic concentrations have been associated with higher mortality by age group. In younger men TEST production could be affected during COVID-19 infection and lead to higher mortality. On the figure, the red line represents TEST plasmatic concentrations, and the blue line illustrates COVID-19 mortality in males.

**Figure 2 ijms-23-00935-f002:**
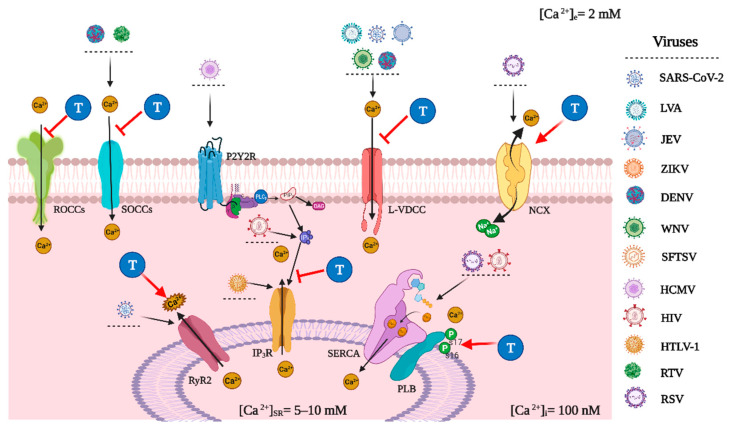
Viral hijacking of Ca^2+^ handling proteins and testosterone modulation. Schematic representation of various stages of the viral cycle targeting the Ca^2+^ apparatus in a host cell. Testosterone (T) can mitigate the dysfunction of Ca^2+^ homeostasis induced by the viral infection through the modulation of the activity and expression of various Ca^2+^ handling proteins. In the plasma membrane, T can acutely inhibit receptor operated calcium channels (ROCCs) in vascular smooth muscle (VSM), T also inhibits store operated calcium channels (SOCCs) acutely in airway smooth muscle (ASM) and VSM. T administered acutely inhibits L-Type voltage operated Ca^2+^ channels (L-VDCCs) in VSM and ASM, and, if chronically given, can downregulate L-VDCCs expression in cardiomyocytes. In these cells, T can upregulate the expression of the Na^+^/Ca^2+^ exchanger (NCX). In the sarcoplasmic reticulum (SR), T can block the IP_3_ receptor (IP_3_R). In cardiomyocytes, chronic exposure to T increases the phosphorylation of phospholamban (PLB) sites s16 and s17, increasing sarcoplasmic reticulum Ca^2+^ ATPase (SERCA) activity, and can also increase the amplitude of Ca^2+^ sparks from the ryanodine receptor (RyR), probably due to an increase in Ca^2+^ content in the SR by the increased SERCA activity. Abbreviations on the figure: LVA, influenza A virus; JEV, Japanese encephalitis virus; ZIKV, Zika virus; DENV, dengue virus; WNV, West Nile virus; SFTSV, thrombocytopenia syndrome virus; HCMV, human cytomegalovirus; HIV, human immunodeficiency virus; HTLV-1, human T-cell lymphotropic virus type 1; RSV, respiratory syncytial virus; RTV, rotavirus; T, testosterone; GPCR, G-protein-coupled receptor; PLC-β, phospholipase C-β; IP_3_, inositol1,4,5-trisphosphate; PIP, phosphatidyl inositol phosphate; DAG, diacylglycerol; b[Ca^2+^]i, basal intracellular Ca^2+^ concentration; [Ca^2+^]SR, sarcoplasmic reticulum Ca^2+^ concentration; [Ca^2+^]e, extracellular Ca^2+^ concentration.

**Figure 3 ijms-23-00935-f003:**
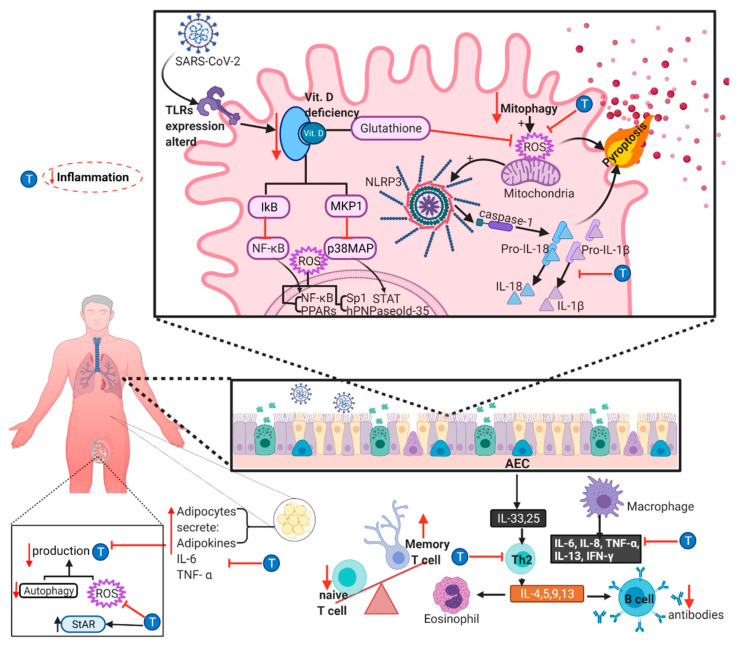
Testosterone mitigates the detrimental effects of inflammaging in COVID-19. Schematic representation of late-onset hypogonadism (LOH) produced by inflammaging markers, reverted with testosterone (T) supplementation, and increasing the steroidogenic acute regulatory protein (StAR) expression leading to T production. Inflammaging is characterized by alterations in autophagy and mitophagy activity, increased reactive oxygen species (ROS) production, cellular senescence, alteration of the expression of toll-like receptors (TLRs) and decrease in the concentration of vitamin D. The higher levels of ROS activate NLRP3, activating IL-1β and IL-18 production and pyroptosis, a mechanism that the StAR could block by inducing T production. ROS also activate the transcription factors hPNPaseold-35, NF-κB, AP-1, Sp1 and PPARs. The senescent adipocytes increase the secretion of adipokines and cytokines, such as IL-6 and TNF-α, that can also be inhibited by T. Immunosenescence can present alteration of the ratio of CD4+/CD8+ T cells, decrease immature T cells, increase memory T cells, alter Th2 response, and modify the production of pro-inflammatory cytokines. Abbreviations on the figure: T, testosterone; hPNPaseold-35, human polynucleotide phosphorylase; AP-1, activator protein 1; Sp1, specification protein 1; PPARs, peroxisomal proliferator-activated receptors; NLRP3, NOD-like receptor 3; Vit. D, vitamin D; ROS, reactive oxygen species; AEC, airway epithelial cells; StAR, steroidogenic acute regulatory protein.

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
