# Peer review of "Could Lower Testosterone in Older Men Explain Higher COVID-19 Morbidity and Mortalities?"

_ijms, 2022, doi:10.3390/ijms23020935_

Round 1

Reviewer 1 Report

Review of "Testosterone in COVID-19, a novel approach".  This review proposes that the rough correlation between severe morbidity and mortalities from COVID-19 (also SARS-CoV and MERS) observed in older men and the decline in testosterone levels might suggest a causal relationship.  Since testosterone is known to play a role in calcium homeostasis and it is known that a number of viruses manipulate calcium levels intracellularly to their advantage, the authors suggest that dysregulation of cellular calcium due to lowered testosterone could lead to conditions that favor increased COVID-19 replication and cellular/organ damage.  Also, since testosterone has an anti-inflammatory activity, lower testosterone levels may result in increased inflammatory tissue damage and more severe disease.  The authors cite 3 references that report low testosterone levels in COVID-19 male patients with most cases a return to normal levels months after infection.  

The authors (and other investigators) reasonably suggest that the correlation between low testosterone levels of elderly men and severity of COVID-19 symptoms is causal and require more research to investigate the mechanism(s).  While there is some information about the role of calcium in COVID-19 infection in cells, more research is needed to understand how cellular regulation/dysregulation of calcium affects viral replication.

I like the review.  The writing is clear.  I am not in love with the title because it is too understated.  One suggestion as a title- "Could lower testosterone in older men explain higher COVID-19 morbidity and mortalities?"  

Author Response

REVIEWER 1

Review of "Testosterone in COVID-19, a novel approach".  This review proposes that the rough correlation between severe morbidity and mortalities from COVID-19 (also SARS-CoV and MERS) observed in older men and the decline in testosterone levels might suggest a causal relationship.  Since testosterone is known to play a role in calcium homeostasis and it is known that a number of viruses manipulate calcium levels intracellularly to their advantage, the authors suggest that dysregulation of cellular calcium due to lowered testosterone could lead to conditions that favor increased COVID-19 replication and cellular/organ damage.  Also, since testosterone has an anti-inflammatory activity, lower testosterone levels may result in increased inflammatory tissue damage and more severe disease.  The authors cite 3 references that report low testosterone levels in COVID-19 male patients with most cases a return to normal levels months after infection. 

The authors (and other investigators) reasonably suggest that the correlation between low testosterone levels of elderly men and severity of COVID-19 symptoms is causal and require more research to investigate the mechanism(s).  While there is some information about the role of calcium in COVID-19 infection in cells, more research is needed to understand how cellular regulation/dysregulation of calcium affects viral replication.

Q: I like the review.  The writing is clear.  I am not in love with the title because it is too understated.  One suggestion as a title- "Could lower testosterone in older men explain higher COVID-19 morbidity and mortalities?"

 A: Authors are grateful for the reviewers comment and changed the manuscripts title in accordance with the reviewer’s suggestion.

Reviewer 2 Report

This is a very interesting article.

regarding this paragraph i would suggest the authors also to specify that the numbers from lines 40-41 are continuously evolving.

Regarding lines 47-48 i would also mention the situation in other countries.

You can find some interesting informations also in this two articles:

DOI 10.2147/RMHP.S284557, DOI 10.3390/microorganisms8111704

Regarding the rest of the article i have no comments just in what concernes the conclusion i would restructure it more concise.

Author Response

REVIEWER 2

This is a very interesting article.

Q1: regarding this paragraph i would suggest the authors also to specify that the numbers from lines 40-41 are continuously evolving.

A: Some words were added in this regard to lines 40-41, updating the data corresponding to the current status and stating that these statistics are continuously being updated.

Q2: Regarding lines 47-48 i would also mention the situation in other countries.

A: Actualized numbers for America and USA were included. Corresponding references were added, and we updated the data corresponding to the current status.

Line number 52-56

According to the Center for Systems Science and Engineering (CSSE) at Johns Hopkins University, in January 2022, confirmed cases in America were 105,416, 916 and USA had the highest incidence [1]. Global Health 50/50 reports that in this country male patient be-tween 50 and 64 years of age presented a death toll almost two times higher than in women of the same age (293.26 vs 170.66 per 100,000, respectively) [3].

Q3: You can find some interesting informations also in this two articles: CITED

DOI 10.2147/RMHP.S284557, DOI 10.3390/microorganisms8111704

A: We thank the reviewer for the advice. Both references were consulted and cited in the text (Lines 42-44, and Lines 76-79, respectively)

Q3: Regarding the rest of the article i have no comments just in what concerns the conclusion i would restructure it more concise.

A: The Conclusions section was shortened.
